# Application of Patient Reported Outcome Measures in Cochlear Implant Patients: Implications for the Design of Specific Rehabilitation Programs

**DOI:** 10.3390/s22228770

**Published:** 2022-11-13

**Authors:** Andrea Frosolini, Giulio Badin, Flavia Sorrentino, Davide Brotto, Nicholas Pessot, Francesco Fantin, Federica Ceschin, Andrea Lovato, Nicola Coppola, Antonio Mancuso, Luca Vedovelli, Gino Marioni, Cosimo de Filippis

**Affiliations:** 1Audiology Unit, Department of Neuroscience DNS, University of Padova, 31100 Treviso, Italy; 2Department of Information Science, University of Milan, 20133 Milan, Italy; 3Unit of Biostatistics, Epidemiology, and Public Health, Department of Cardiac, Thoracic, Vascular Sciences, and Public Health, University of Padova, 35100 Padova, Italy; 4Otolaryngology Section, Department of Neuroscience DNS, University of Padova, 35100 Padova, Italy

**Keywords:** cochlear implant, hearing loss, music rehabilitation, PRO measures, NCIQ, MuRQoL, MUSQUAV

## Abstract

Introduction: Cochlear implants (CI) have been developed to enable satisfying verbal communication, while music perception has remained in the background in both the research and technological development, thus making CI users dissatisfied by the experience of listening to music. Indications for clinicians to test and train music abilities are at a preliminary stage compared to the existing and well-established hearing and speech rehabilitation programs. The main aim of the present study was to test the utility of the application of two different patient reporting outcome (PRO) measures in a group of CI users. A secondary objective was to identify items capable of driving the indication and design specific music rehabilitation programs for CI patients. Materials and Methods: A consecutive series of 73 CI patients referred to the Audiology Unit, University of Padova, was enrolled from November 2021 to May 2022 and evaluated with the audiological battery test and PRO measures: Musica e Qualità della Vita (MUSQUAV) and Nijmegen Cochlear Implant Questionnaire (NCIQ) Italian version. Results: The reliability analysis showed good consistency between the different PRO measures (Cronbach’s alpha = 0.873). After accounting for the epidemiological and clinical variables, the PRO measures showed a correlation with audiological outcomes in only one case (rho = −0.304; adj. *p* = 0.039) for NCIQ-T with the CI-pure tone average. A willingness for musical rehabilitation was present in 63% of patients (Rehab Factor, mean value of 0.791 ± 0.675). Conclusions: We support the role of the application of MUSQUAV and NCIQ to improve the clinical and audiological evaluation of CI patients. Moreover, we proposed a derivative item, called the rehab factor, which could be used in clinical practice and future studies to clarify the indication and priority of specific music rehabilitation programs.

## 1. Introduction

The bionic ear, the gold standard treatment for profound hearing loss [1], has improved the hearing functionality, communication skills, and social lives of millions of people around the world in the past decades [2,3]. Although cochlear implants (CIs) have been developed to enable a satisfying verbal communication [4,5], music experiences have remained in the background of research and technological development, and consequently, many CI users are dissatisfied by the music they listen to [6,7]. Despite this fact being widely known by clinicians and patients, a recent systematic review concluded that currently, no single test has been widely used, in a research or clinical context, to assess music experience after cochlear implantation [8,9]. This is reasonable, considering the complexity and dynamicity of music experiences in everyday life. Moreover, the differences in acoustic discrimination and cognitive resources available in hearing impaired patients for managing such complex listening environments have to be taken into account [10,11,12]. It has been previously shown that CI users have scarce tone sensitivity and subsequent problems with melodic identification and timbre discrimination [13]. Nonetheless, the recognition of rhythm and the emotions of music have been reported to be closer to normal hearing subjects [12,14] and there is no strong or constant relation between individual musical performance and musical appraisal [15,16].

Patient reported outcomes (PROs) are increasingly used to assess, with standardized methods, a range of outcomes including symptoms, functional health, well-being, and psychological issues from the patients’ perspective [17]. Among these instruments, the Nijmegen Cochlear Implant Questionnaire has been applied in different populations to evaluate both specific and general functional outcomes [18,19], while the Music Related Quality of Life questionnaire was definitively developed to test music perception and the engagement of CI patients in several real-life contexts. The original author of the MuRQoL questionnaire stated that, considering the pervasive presence of music in daily life and its role in emotional expression and cultural connection, musical perception and participation could have a correlation with quality of life [20]; our research group has previously translated and cross-culturally validated this instrument into an Italian version [20,21].

Music experience has been evaluated in different subgroups of CI patients such as preverbal pediatric patients, post-verbal, pre-verbal lately implanted, bilateral users, bimodal users, and unilateral users [10,22,23]. These different populations have reported heterogeneous rehabilitation needs, but clear indications are lacking for clinicians to develop and prescribe music rehabilitation programs for CI users [6,24,25]. Some open questions, at the actual state-of-the-art, need to be addressed: should clinicians try to improve music perception [26,27] or rather focus on eliciting an equivalent emotional response to music [28]? How can we bring research toward more ecological, real-life-like situations and how should a clinician indicate dedicating time for music when resources are already severely limited for speech focused interventions [6]?

The main objective of the present study was to test the utility of the application of two different PRO measures in a group of CI users. A secondary aim was to identify items capable of driving the indication and design of specific music rehabilitation programs for CI patients.

## 2. Materials and Methods

### 2.1. Study Design and Ethical Approval

This mixed embedded study, composed of a survey combined with a retrospective data collection, was conducted in accordance with the principles of the Declaration of Helsinki [29]. Data were examined in compliance with Italian privacy and sensitive data laws, and with the in-house rules of our institution. Informed consent was obtained from each participant. Ethical approval was obtained by the local committee (“Comitato Etico Marca Trevigiana” number 1196/CE).

### 2.2. Participants

A consecutive cohort of CI patients referred to the Audiology Unit, Treviso Hospital, Neuroscience Department, University of Padova, was enrolled from November 2021 to May 2022.

Inclusion criteria were as follows:Patients older than 11 years;Last CI surgery at least 12 months before evaluation;Regular follow-up controls.

Exclusion criteria were: Not willing to complete the survey;Presence of neurological or psychiatric disorders.

The CI procedure was performed for the vast majority of patients by the expert surgeon of the group (CdF) using a posterior tympanotomy approach optimized to enhance hearing preservation [3,30].

The following demographic and clinical data were recorded: age, gender, temporal onset and etiological classification of deafness, years of hearing deprivation, years of CI use, linguistic, and musical skills.

### 2.3. PRO Measures

The Italian Nijmegen Cochlear Implant Questionnaire (I-NCIQ) and the Musica e Qualità della Vita (MUSQUAV) questionnaires [21] (Italian translation of Music related Quality of Life [20]) were administered to the enrolled patients. Both tests are based on a 5-point Likert scale.

The MUSQUAV questionnaire is a novel instrument for the assessment of the patients’ perception and musical engagement with the chance to give specific indications for rehabilitation programs. The questionnaire consists of two mirror sections, each one containing 18 questions divided into two subsections. The first section, named “frequency”, analyzes how often the subject is able to perceive and be engaged in music, whereas the second section, “importance”, examines how relevant the listening of music is for the subject and their engagement in the listening itself. The subsections are, indeed, perception (questions 1–11) and engagement (questions 12–18) [21].

The I-NCIQ is a widely used instrument designed to quantify the quality of life in patients with CIs [31]. It is composed of six different sub-domains: basic sound perception; advanced sound perception; speech production; self-esteem; activity limitations; and social interactions. The time needed to complete the two surveys is approximately 20 min. The patients manually filled in and answered the questionnaires [21,31]; the data were then acquired in an Excel spreadsheet (Microsoft Excel 2019 for Windows 10) by researchers of our group.

### 2.4. Developing of the Rehab Factor

In order to propose a numerical factor to quantify the individual musical rehabilitation needs, the difference between the frequency score and importance score, as expressed in the two sections in the MUSQUAV (MUSQUAV Importance and MUSQUAV frequency), was calculated for each patient. Patients who had values of importance less than two out of five on the Likert scale (“not at all relevant” or “not very relevant”) were excluded. The value obtained, by definition greater than 0, was called the rehab factor.

### 2.5. Audiological Evaluation

Audiological results at last evaluation (within 12 months before the day of the observation) were considered for each patient. Audiometry was performed with a Madsen Astera by GN Otometrics (Taastrup, Denmark), in accordance with European (IEC 60645-I) and ISO (389-1) standards, in an audiometric test booth. We tested the hearing thresholds without hearing devices and hearing thresholds and speech audiometry with hearing devices in the best-aided condition. The pure tone average (PTA2, considering threshold levels at 0.5, 1, 2, and 4 kHz), the speech reception threshold and speech intelligibility threshold (SRT and SIT, respectively, the intensity in decibels at which 50% and 100% of a disyllabic word were recognized) were considered, as previously reported [21].

### 2.6. Statistical Analysis

Reliability analysis was conducted to test the correlation between MUSQUAV and NCIQ, the Cronbach’s alpha value was calculated, and the Pearson correlation heat map was reported.

Correlation and partial correlation analyses were conducted using the Spearman test for PRO measures and all the previously cited demographical, clinical, and audiological variables. For all measures, all p values were corrected for the false discovery rate and significance was set at adj. *p* < 0.05.

The jamovi software (version 1.6, 2021, open access software available at https://www.jamovi.org, accessed on 1 September 2022) was used for our statistical purposes [32].

## 3. Results

### 3.1. Group Data

Seventy-three patients were included, 46 females (63%) and 27 males (37%); Table 1 summarizes the main demographic, clinical, and audiological characteristics reporting the mean values, median, standard deviation, interquartile range, and range. The main hearing loss etiologies (genetic, infective, autoimmune, and idiopathic) were present. The onset of hearing loss was slightly predominantly post-verbal (40 cases, 54.8% vs. 45.2%). Rehabilitation strategies were distributed between unilateral CI (29 cases, 39.7%), bilateral CI (21 cases, 28.8%), and bimodal rehabilitation CI (23 cases, 31.5%).

### 3.2. PRO Measures Correlations

The reliability analysis showed good consistency between the different PRO measures (MUSQUAV and NCIQ), with a Cronbach’s alpha value of 0.873. The Pearson’s test showed a significant positive correlation between the frequency section of the MUSQUAV and the total NCIQ (r = 0.632; *p* < 0.001) as well as all its subdomains, with the exception of NCIQ2 (enhanced sound perception), as depicted in Figure 1. The correlation of the importance section of the MUSQUAV was weaker with the total NCIQ (r = 0.246; *p* = 0.036) and only with subdomains NCIQ1 (r = 0.277; *p* = 0.018) and NCIQ3 (r = 0.425; *p* < 0.001). The expected correlations between subdomains of the NCIQ are reported in Figure 1.

Spearman’s correlation for PRO measures with epidemiological, clinical, and audiological variables found that age (rho = −0.399, adj. *p* = 0.020), time of onset of hearing loss (rho = −0.367, adj. *p* = 0.016), CI-PTA (rho = −0.292, adj. *p* = 0.033), SRT (rho = −0.365, adj. *p* = 0.007), SIT (rho = −0.427, adj. *p* = 0.011) were all negatively correlated with the MUSQUAV frequency. NCIQ-T was negatively correlated with CI-PTA (rho = −0.352, adj. *p* = 0.009). Other correlations between the demographic, clinical, and audiological variables are shown in Table 2. We further conducted a partial regression including epidemiological and clinical data as the control variables for the PRO measures and audiological outcomes. The correlations were confirmed for F-MUSQUAV with I-MUSQUAV (rho = 0.447, adj. *p* = 0.015) and NCIQ-T (rho = 0.582, adj. *p* = 0.008) and for NCIQ-T with CI-PTA (rho = −0.304, adj. *p* = 0.039). The correlation between the following audiological measures was also confirmed: CI-PTA with SRT (rho = 0.520, adj. *p* = 0.005); CI-PTA with SIT (rho = 0.403, adj. *p* = 0.040); SRT with SIT (rho = 0.540, adj. *p* = 0.004), as expected (see Table 3).

### 3.3. The REEHAB Factor

The rehab factor was present for 46 out of 73 patients (63%), with a mean value of 0.791 ± 0.675 (range 0.033–2.44). The remaining 26 out of 73 patients had an I-MUSQUAV score lower than 2/5 and/or F-MUSQUAV score greater than the I-MUSQUAV score. Descriptive values of the patients with rehab factors are reported in Table 4 and Table 5.

The partial correlation for audiological outcome and NCIQ scores, considering the epidemiological and clinical data as control variables, showed a positive correlation of REHAB with SRT (rho = 0.417, adj. *p* = 0.050) and a negative correlation with NCIQ-3 (rho = −0.570, adj. *p* = 0.01), NCIQ3 (rho = −0.570, adj. *p* = 0.010) and NCIQ4 (rho = −0.344, adj. *p* = 0.080) (see Table 6).

## 4. Discussion

In recent years, after achieving strong, stable, and amazing results with verbal communication [33,34], CI research has focused much more on how to improve music listening and the participation of implanted patients [35].

In this original research, we tried to assess some of these questions by examining 73 consecutive patients presenting to a tertiary referral center for audiological and phoniatric diseases, expanding the field of knowledge of our precedent investigation on 180 normal hearing subjects and 35 post-verbal CI patients [21]. Due to unrestricted inclusion criteria, the study group had a wide age distribution, different etiologies, time of onset of hearing loss, hearing rehabilitation strategies, and CI experience, as summarized in Table 1. CI-PTA (29.8 ± 5.84 dB) and SRT (40.6 ± 9.27 dB) tests, as expected, revealed auditory performances adequate to ensure good verbal perception in most patients. These good audiological outcomes consequently had a positive impact on the quality of life, as confirmed by the average scores of 3.59 ± 0.524 at the I-NCIQ. Nonetheless, the I-NCIQ scores showed considerable variability (from 2.05 to 4.66), justifiable by the heterogeneity of a consecutive group of patients. Accordingly, the scores of the MUSQUAV showed marked variability: the mean F-MUSQAV was 3.00 ± 0.864 (median 3.11, range 1.50–4.65), consistent with a self-rating of musical abilities in the study group that was overall adequate for individual expectations but inferior to the median value of 3.94 previously found in a group of 97 normal hearing subjects [21]. The mean importance section of MUSQUAV was 3.34 ± 0.798 (median 3.44, range 1.50–5.00). The higher value of I-MUSQUAV section in comparison with the F-MUSQUAV section indicates a frequent discrepancy between the self-evaluation of musical abilities/engagement (F-MUSQUAV) and the individually rated importance of such properties (I-MUSQUAV). This is typical of an impaired hearing group and absent in normal hearing subjects, as previously reported [21].

We chose MUSQUAV, the Italian translation of The Music related Quality of Life questionnaire [20], to test the music perception and engagement of CI patients in several real-life contexts, since no standard of evaluation of music perception is available to date [8]. Following the results of the present research, we can support the relationship between MUSQUAV and the quality of life claimed by the original authors of the questionnaire [20] due to the moderate-strong positive association revealed in our sample between the F-MUSQUAV and NCIQ scores (r = 0.632, *p* < 0.001) at the Pearson’s test. Moreover, the factor analysis of MUSQUAV and NCIQ items resulting in having a Cronbach’s alpha value higher than 0.8, which can be interpreted as a relevant indicator of the external consistency of the MUSQUAV questionnaire, never before tested, to the best of our knowledge (see Figure 1). Moreover, in analogy with NCIQ’s previous utilization in pediatric populations [36], in this investigation, we successfully applied the MUSQUAV questionnaire to adolescent patients (age greater than 11 years old), for the first time to the best of our knowledge.

We also aimed to investigate the associations between the PRO measures and clinical/audiological outcomes. Since several correlations were observed at an exploratory analysis, as expected, with age, hearing loss onset, and CI-use (Table 2), we further conducted a partial correlation using epidemiological and clinical variables as controlling factors (Table 3). The already reported association between F-MUSQUAV, I-MUSQUAV, and NCIQ was confirmed in both analyses; only the association between PRO measures and audiological outcome was found for NCIQ and CI-PTA, revealing a weak negative correlation (rho = −0.304, Table 3). These results are in line with those reported by Vasil et al. [18], who considered a group of 44 CI users and found that NCIQ had no correlation with standard audiological outcome, concluding that clinicians might integrate information obtained by PRO measures to better estimate the real-world performance of CI patients and improved counseling and the development of recommendations [18]. Accordingly, in response to our primary research question, our data support the hypothesis that validated PRO measures such as MUSQUAV and NCIQ may be applied in the context of CI clinics to test the abilities and weaknesses that go unnoticed in standard audiology battery tests, with the purpose of giving a better indication to rehabilitation programs.

The most important novelty and unique feature in this study is related to the introduction of a quantitative item for the analysis of individual rehabilitation needs. The rehab factor was determined by the difference between MUSQUAV importance and frequency, in other words, the delta between the importance given to music and the executable skills and activities in the field of music, as expressed by individual subjects when answering the two specular sections of the MUSQUAV questionnaire [21]. The rehab factor is not valid if the subject gives no or scarce importance to music, which happens when the I-MUSQUAV score is lower than or equal to 2 on the 5-point Likert scale. The rehab factor was present in 46 out of 73 patients (63%), with a mean value of 0.791 ± 0.675 (median 0.514, range 0.032–2.44). It follows that most patients referred to a CI clinic could require a direct rehabilitative intervention in various areas of the musical experience. This confirms data previously reported, in which even 90% of CI users were subjectively wishing to undergo a music rehabilitation program [10]. In the partial correlation, the rehab factor showed a positive correlation with SRT (rho = 0.417, adj. *p* = 0.05) and a negative one with NCIQ3 (rho = −0.570, *p* = 0.01). Therefore, the REHAB factor correlates with weaker audiological performances (higher SRT values) and poor self-rated outcomes (lower NCIQ scores), data that can preliminarily suggest an ability of the rehab factor to intercept rehabilitation needs within the patient group.

The fact that the difference between importance and frequency could be proportional to the impact on quality of life was reported by the authors of the questionnaire as well as the possibility to plot individual data in a matrix to draw rehabilitation programs [20], but to the best of our knowledge, a numerical factor has not been previously proposed by any other research group. In answer to the second objective of our study, the rehab factor could be proposed in clinical practice after the verification, in future research, of the power and validity measures of the test such as sensitivity, specificity, positive predictive, and negative predictive values.

The rationale of the inclusion of music programs in the rehabilitation course of CI patients takes place in the recently emerging concepts of cross-modal plasticity and multisensory integration in hearing impaired subjects [37,38]. These latter pieces of evidence support the future application of polimodal rehabilitation strategies. Taking into account the limits of the available auditory programs evidenced in recent systematic reviews [39], researchers are conducting clinical trial protocols and randomized control trials to test the efficacy of integrative rehabilitation methods [40,41].

The limits of this work were mainly related to the single center design of the study. Moreover, the data collection was undertaken during the COVID-19 pandemic period, and this could have decreased the score of some items of PRO measures, especially considering that pandemic restrictions had a negative impact on individual musical activities [42].

## 5. Conclusions

The preliminary results of the present research support the role of the application of two different PRO measures (MUSQUAV and NCIQ) to improve the clinical and audiological evaluation of CI patients. Moreover, we proposed a derivative item (the REHAB Factor) which, after verification of its statistical power in future research projects, could be used in clinical practice to clarify the indication and priority of specific music rehabilitation programs for CI patients.

## Figures and Tables

**Figure 1 sensors-22-08770-f001:**
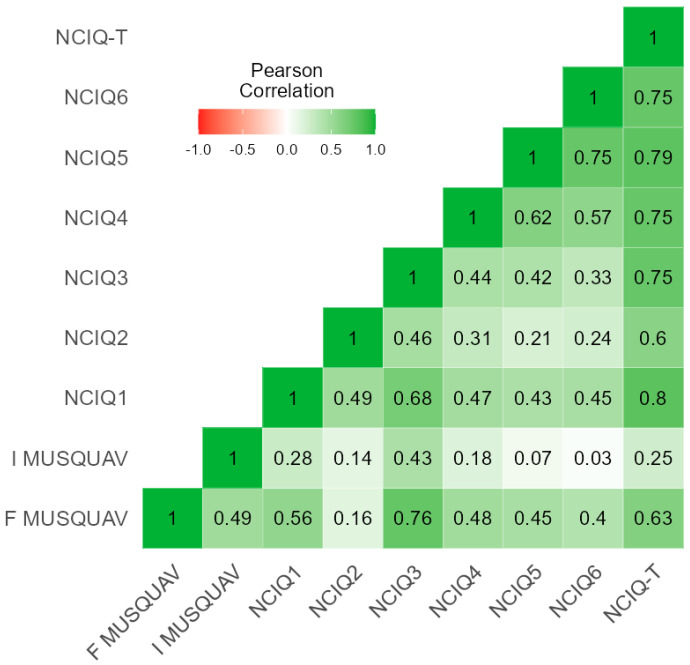
Correlation heat map of different PRO measures: NCIQ and MUSQUAV.

**Table 1 sensors-22-08770-t001:** Clinical features, audiological outcome, and PRO measures of the included participants.

Descriptives	N	Missing	Mean	Median	SD	IQR	Minimum	Maximum
Age (years)	73	0	47.1	49.0	23.1	46.0	11.0	89.0
CI use (years)	73	0	9.75	10.0	6.32	10.00	1.00	27.00
Auditory deprivation (years)	73	0	7.32	0.00	11.8	13.00	0.00	65.00
PTA (dB)	73	0	104	115	22.6	23.8	35.0	120
CI-PTA (dB)	73	0	29.8	30.0	5.84	8.75	20.0	50.0
SRT (dB)	69	4	40.6	40.0	9.27	10.0	22.0	63.0
SIT (dB)	42	31	51.9	50.0	9.94	17.5	40.0	70.0
F MUSQUAV	73	0	3.00	3.11	0.864	1.46	1.50	4.65
I MUSQUAV	73	0	3.34	3.44	0.798	0.83	1.50	5.00
REHAB	46	27	0.791	0.514	0.675	0.903	0.033	2.44
NCIQ1	73	0	3.59	3.60	0.770	1.20	1.60	5.00
NCIQ2	73	0	4.04	4.10	0.675	1.00	2.29	5.00
NCIQ3	73	0	3.44	3.50	0.664	1.10	1.70	4.60
NCIQ4	73	0	3.35	3.30	0.563	0.700	2.20	4.60
NCIQ5	73	0	3.66	3.70	0.861	1.30	1.30	5.00
NCIQ6	73	0	3.47	3.57	0.658	1.00	1.78	4.71
NCIQ-T	73	0	3.59	3.60	0.524	0.733	2.05	4.66

Abbreviations: F MUSQUAV (Frequency questionnaire of Music and Quality of Life questionnaire); I MUSQUAV (Importance section of Music and Quality of Life questionnaire); IQR (interquartile range); N (number of subjects); NCIQ-T (Nijmegen Cochlear Implant Questionnaire Total); NCIQ 1,2,3,4,5,6 (Subsections of Nijmegen Cochlear Implant Questionnaire); SD (standard deviation); SRT (speech recognition threshold); SIT (speech intelligibility threshold); CI-PTA (pure tone average with cochlear implant); PTA (pure tone average); CI (cochlear implant).

**Table 2 sensors-22-08770-t002:** Correlation matrix for PRO measures and both the clinical characteristics and audiological outcome.

		F MUSQUAV	I MUSQUAV	NCIQ-T	Gender	Age	Onset	Rehabil.	CI Use	Aud. Depr.	PTA	CI-PTA	SRT	SIT
**F MUSQUAV**	ρ	-												
	adj. *p*-v.	-												
**I MUSQUAV**	ρ	**0.488**	-											
	adj. *p*-v.	**0.008 ***	-											
**NCIQ-T**	ρ	**0.609**	0.209	-										
	adj. *p*-v.	**0.039 ***	0.132	-										
**Gender**	ρ	0.197	0.049	0.083	-									
	adj. *p*-v.	0.143	0.717	0.557	-									
**Age**	ρ	**−0.399**	**−0.300**	−0.223	−0.131	-								
	adj. *p*-v.	**0.020 ***	**0.029 ***	0.103	0.334	-								
**Onset**	ρ	**−0.367**	−0.204	−0.133	−0.045	**0.601**	-							
	adj. *p*-v.	**0.016 ***	0.137	0.328	0.732	**0.005 ***	-							
**Rehabil.**	ρ	−0.128	−0.204	0.001	−0.201	**0.293**	0.270	-						
	adj. *p*-v.	0.342	0.138	0.990	0.142	**0.032 ***	0.051	-						
**CI use**	ρ	0.261	0.112	0.127	0.151	−0.247	**−0.421**	**−0.598**	-					
	adj. *p*-v.	0.058	0.403	0.342	0.274	0.072	**0.007 ***	**0.006 ***	-					
**Aud. depr.**	ρ	0.178	0.148	0.050	0.242	−0.160	−0.249	**−0.845**	**0.630**	-				
	adj. *p*-v.	0.198	0.273	0.718	0.076	0.239	0.074	**0.009 ***	**0.010 ***	-				
**PTA**	ρ	−0.162	0.040	−0.065	−0.077	−0.178	−0.043	**−0.417**	**0.277**	0.074	-			
	adj. *p*-v.	0.243	0.750	0.633	0.579	0.192	0.740	**0.004 ***	**0.045 ***	0.586	-			
**CI-PTA**	ρ	**−0.292**	−0.178	**−0.352**	−0.370	**0.428**	0.137	**0.270**	−0.233	−0.197	−0.117	-		
	adj. *p*-v.	**0.033 ***	0.194	**0.009 ***	0.078	**0.005 ***	0.315	**0.049 ***	0.089	0.147	0.385	-		
**SRT**	ρ	**−0.365**	−0.166	−0.270	−0.255	**0.659**	**0.384**	**0.372**	**−0.314**	−0.233	−0.250	**0.682**	-	
	adj. *p*-v.	**0.007 ***	0.240	0.057	0.072	**0.006 ***	**0.013 ***	**0.007 ***	**0.027 ***	0.098	0.076	**0.004 ***	-	
**SIT**	ρ	**−0.427**	−0.106	−0.199	**−0.377**	**0.731**	**0.476**	**0.447**	**−0.448**	−0.306	−0.276	**0.728**	**0.844**	-
	adj. *p*-v.	**0.016 ***	0.572	0.272	**0.036 ***	**0.007 ***	**0.016 ***	**0.010 ***	**0.010 ***	0.091	0.131	**0.010 ***	**0.005 ***	-

Note: Controlling for ‘Gender’, ‘Age’, ‘Onset’, ‘Rehabilitation’, ‘CI use’, ‘Auditory deprivation’, and ‘PTA’. * = Significant using a false discovery rate of 0.05. Abbreviations: adj. *p*-v. (adjusted *p*-value); aud. depr. (auditory deprivation); F MUSQUAV (Frequency questionnaire of Music and Quality of Life questionnaire); I MUSQUAV (Importance section of Music and Quality of Life questionnaire); N (number of subjects); NCIQ-T (Nijmegen Cochlear Implant Questionnaire Total); Rehabil. (rehabilitation); SRT (speech recognition threshold); SIT (speech intelligibility threshold); CI-PTA (pure tone average with cochlear implant); PTA (pure tone average); CI (cochlear implant); ρ (Spearman’s rho).

**Table 3 sensors-22-08770-t003:** Partial correlation matrix for PRO measures and both the clinical characteristics and audiological outcome.

		F MUSQUAV	I MUSQUAV	NCIQ-T	CI-PTA	SRT	SIT
**F MUSQUAV**	ρ	-					
	adj. *p*-v.	-					
**I MUSQUAV**	ρ	**0.447**	-				
	adj. *p*-v.	**0.015 ***	-				
**NCIQ-T**	ρ	**0.582**	0.171	-			
	adj. *p*-v.	**0.008 ***	0.364	-			
**CI-PTA**	ρ	−0.124	−0.051	**−0.304**	-		
	adj. *p*-v.	0.401	0.687	**0.039 ***	-		
**SRT**	ρ	−0.084	0.125	−0.146	**0.520**	-	
	adj. *p*-v.	0.555	0.385	0.387	**0.005 ***	-	
**SIT**	ρ	−0.180	0.225	−0.224	**0.403**	**0.540**	-
	adj. *p*-v.	0.412	0.362	0.327	**0.040 ***	**0.004 ***	-

* = Significant using a false discovery rate of 0.05. Abbreviations: adj. *p*-v. (adjusted *p*-value); aud. depr. (auditory deprivation); F MUSQUAV (Frequency questionnaire of Music and Quality of Life questionnaire); I MUSQUAV (Importance section of Music and Quality of Life questionnaire); NCIQ-T (Nijmegen Cochlear Implant Questionnaire Total); SRT (speech recognition threshold); SIT (speech intelligibility threshold); CI-PTA (pure tone average with cochlear implant); ρ (Spearman’s rho).

**Table 4 sensors-22-08770-t004:** Demographic, clinical characteristics, audiological outcomes, and PRO measures of the Descriptives—REHAB group.

Descriptives	N	Missing	Mean	Median	SD	IQR	Minimum	Maximum
Age (years)	46	0	46.9	47.0	21.3	34.3	11.0	79.0
CI use (years)	46	0	9.57	9.00	6.59	10.0	1.00	27.0
Auditory depr. (years)	46	0	6.96	0.00	10.7	13.0	0.00	48.0
PTA (dB)	46	0	107	116	19.2	19.1	48.8	120
CI-PTA (dB)	46	0	30.3	30.0	5.79	9.69	21.3	50.0
SRT (dB)	44	2	41.0	40.0	6.77	7.50	25.0	57.0
SIT (dB)	23	23	54.3	50.0	8.96	10.0	40.0	70.0
F MUSQUAV	46	0	2.85	2.79	0.75	1.15	1.61	4.39
I MUSQUAV	46	0	3.64	3.56	0.59	0.74	2.50	5.00
REHAB	46	0	0.791	0.514	0.675	0.903	0.033	2.44
NCIQ1	46	0	3.50	3.60	0.703	0.975	1.60	4.56
NCIQ2	46	0	4.05	4.01	0.700	1.09	2.29	5.00
NCIQ3	46	0	3.39	3.40	0.625	1.00	1.70	4.60
NCIQ4	46	0	3.31	3.20	0.564	0.667	2.30	4.60
NCIQ5	46	0	3.50	3.58	0.870	1.28	1.30	4.80
NCIQ6	46	0	3.33	3.40	0.687	1.04	1.78	4.44
NCIQ-T	46	0	3.51	3.55	0.526	0.763	2.05	4.59

Abbreviations: F MUSQUAV (Frequency questionnaire of Music and Quality of Life questionnaire); IQR (interquartile range); I MUSQUAV (Importance section of Music and Quality of Life questionnaire); NCIQ-T (Nijmegen Cochlear Implant Questionnaire Total); NCIQ 1,2,3,4,5,6 (Subsections of Nijmegen Cochlear Implant Questionnaire); SD (standard deviation); SRT (speech recognition threshold); SIT (speech intelligibility threshold); CI-PTA (pure tone average with cochlear implant); PTA (pure tone average); CI (cochlear implant).

**Table 5 sensors-22-08770-t005:** Demographic, clinical characteristics, audiological outcomes, and PRO measures of the Frequencies of Gender, Onset, and Rehabilitation—REHAB group.

Variables	N	% of Total
**Gender**		
Female	33	71.7%
Male	13	28.3%
**Onset**		
Pre-verbal	19	41.3%
Post-verbal	27	58.7%
**Rehabilitation**		
Unilateral CI	18	39.1%
Bilateral CI	15	32.6%
Bimodal	13	28.3%

Abbreviations: CI (cochlear implant).

**Table 6 sensors-22-08770-t006:** Partial correlation matrix of the rehab factor with the clinical characteristics, audiological outcome, and NCIQ.

	CI-PTA	SRT	SIT	NCIQ-T	NCIQ1	NCIQ2	NCIQ3	NCIQ4	NCIQ5	NCIQ6
REHAB										
Spearman’s rho	0.221	0.417	0.067	−0.356	−0.185	−0.092	−0.570	−0.344	−0.203	−0.230
adj. *p*-value	0.293	0.050 *	0.806	0.087	0.325	0.642	0.010 *	0.080	0.307	0.318

* = Significant using a false discovery rate of 0.05. Abbreviations: NCIQ-T (Nijmegen Cochlear Implant Questionnaire Total); NCIQ 1,2,3,4,5,6 (Subsections of Nijmegen Cochlear Implant Questionnaire); SRT (speech recognition threshold); SIT (speech intelligibility threshold); CI-PTA (pure tone average with cochlear implant).

## Data Availability

The data that support the findings are stored on the online repository https://researchdata.cab.unipd.it/; https://doi.org/10.25430/researchdata.cab.unipd.it.00000713 (accessed on 30 September 2022).

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
