# Peer review of "Application of Patient Reported Outcome Measures in Cochlear Implant Patients: Implications for the Design of Specific Rehabilitation Programs"

_sensors, 2022, doi:10.3390/s22228770_

Round 1

Reviewer 1 Report

The authors handle the rehabilitative importance of patient reported outcome measure under the aspect of music perception and a CI specific validated questionaire. The manuscript is wrongly submitted to a sensor specific journal and not a rehabilitatvie orientated journal.

Author Response

Thank you for considering our manuscript. The rationale for the submission of this manuscript to the section "State-of-the-Art Sensors Technologies", Special Issue "State-of-the-Art Sensors Technology in Italy 2022” was mainly related to the evidence that Cochlear Implants, the first examples of an artificial replacement of a human sense, are often lacking in reproducing advanced sound scenarios such as music. In the context of deaf patients’ management, the development of a “clinical sensor” derived from PRO measures (e.g. the clinical item we proposed and called REHAB FACTOR), which is the scope of the submitted article, could be of interest for the scientific community and the readers of the journal SENSORS. All these theoretical passages, implied in this investigation, have been further highlighted in this revised version.

Reviewer 2 Report

The paper "Application of Patients Reported Outcome measures in Cochlear Implant patients: implications for the design of specific rehabilitation programs" adresses a very interesting and important issue - assessment  of QoL in CI patients and more specific the part of music QoL.

The authors state: "The main objective of the present study was to test the utility of the application of two different PRO measures in a group of CI users" (Introduction). The utility of the NCIQ is not novel but wide spread. The novelty seems to be the MUSQUAV, i.e. an aspect of QoL concerning musical aspects. Since the authores did conduct a previous study on MUSQUAV (reference 22) it remains unclear why they need to to a utility check - and weather the outline of the current study is feasible to answer this question.

The authores invented a simple tool, the "REHAB factor" that might give both cliniciansand administration a ground to decide on therapy options in the CI rehabilitation. I would strongly suggest to focus on this novel aspect and to omit large parts of correlation analyses (especially since the significance of the results is strongly limited by the very large number of correlations).

Unfortunately the paper has several flaws. First an extensive editing of English language and style required.

The paper should be restructured at some points:

Abstract:

The authors state, that "precise indications for clinicians to test music abilities... are lacking". This is not true! 

Introduction

In the introduction, the authors give a number of "hints" concerning their methodology (e.g. the choice of the specific questionnaire or the role of music in CI-lives) - but authors owe an explanation. These explanations are insterded into the discussion section. I would suggest to remove the first and third paragraph of section 4 Discussion to section 1 Introduction. Apart from that, the authors are required to add much more information on the current state of the art concerning music perception - both music accuracy (can the CI users detect musical elements?) and appraisal (do the CI users enjoy what they hear?) - both aspects are independent factors though appraisal is much more important for QoL.

The authors focus on two questionnaires - the NCIQ and the Music Related Quality of Life Questionnaire. On further reading, it can be seen that the group of authors has obviously already worked on or with this MuRQoL. However, this only becomes apparent upon closer reading - much ambiguity could be avoided here if the preliminary work were directly addressed and the current research project were presented as a logical consequence of the previous work. Then the suspicion of an unseemly self-citation would not arise.

Abbreviations: The authors use a multiplicity of abbreviations and acronyms. It is rather tedious to comprehend the tables / figures. Please simplify your presentation - by drastically reducing the amount of data presented or by including a centralized list of abbreviations (I would prefer the latter solution).

Methods

Participants --> Description of inclusion/exclusion criteria is clear, but why did you chose to mix "adult" and "child" population? Are the questionnaires suitable / approved for underage CI-users? 

PRO-Measures --> explain the questionnaires in  more detail. Why did you use the MUSQUAV? It is not possible to follow your arguments wihtout background knowledge on this questionnaire. In the results you refer to differen subscales of both the MUSQUAV and the NCIQ but these are not mentioned in the methods and remain as such rather incomprehensible. 

Developing of the Rehab Factor --> What do the subscale Frequency and Importance mean? Please explain!

Statistical Analysis --> you explain, that "...for all measures, a multiplicity test to rule out False discovery rate was run and an alpha of 0.05 was set: when p > 0.05, correlations are reported as not significant". Given the multiplicity of correlations it is not clear how a alpha of 0.05 can achieve this goal - please explain or adjust the alpha accordingly. 

Preferably you should consider

a) using different statistical methods, e.g. regression analysis taking into account the different subscales

b) reducing the amount of variables

Results

The results are rather tedious to read given the multiplicity of descriptive data and correlation. I would strongly recommend the authors to simplify the data selection and data presentation significantly and to focus on the essential results.

Table 1a ant Table 4a: Explain abbreviations (SD, IQR, NCIQ & MUSQUAV Subscales); 

Table 1b and Table 4b: At least Cumulative % can be omitted (or cancel table 1b completely, data is presented in the text)

Figure 1: Interesting and fast overview though data are presented three times: in the figure, in the tables and in the text. Please consider simplification!

Table 2: This table is not referred to in the - either explain the data or omit table 2

Discussion

The authors refer to their own previous work on the MUSQUAV - it would be far better, to emphasize this previous work and to present the objective of the current study as a continuation of the first implenentation study.

The unique feature of the article is the "Rehab Factor" and this should be emphazised in the discussion. Please explain the "Rehab Factor" in much more detail - what does a mean value of 0,73 mean? Consider an easily comprehensible value (e.g. 0-100%)

Unfortunately, the results are not discussed in the context of the extensive research on music perception and experience in CI users - here much publicity could be given to an extension of conventional rehabilitation to include the aspect of music (in relation to quality of life but also in relation to non-verbal parts in everyday communication)

Author Response

Dear reviewer, thanks for reviewing our manuscript. Please find attached our point-to point response.

Reviewer 3 Report

The authors wrote an article about the Application of Patients Reported Outcome measures in Cochlear Implant patients and the implications for the design of specific rehabilitation programs, regarding the music perception. The article is very interesting, well written and the topic is very hot. I have some suggestions to improve the scientific quality of the article and to give a better contribution in literature.

1. In the statistical analysis is useful to verify the normality of values using the kolmorog-smirnov test.

2. Please in the methods describe the kind of surgery for CI. Because soft surgery (round window approach) gives better outcomes than cocleostomy. Use these references: Freni F, Gazia F, Slavutsky V, Scherdel EP, Nicenboim L, Posada R, Portelli D, Galletti B, Galletti F. Cochlear Implant Surgery: Endomeatal Approach versus Posterior Tympanotomy. Int J Environ Res Public Health. 2020 Jun 12;17(12):4187.    and     Tarabichi O, Jensen M, Hansen MR. Advances in hearing preservation in cochlear implant surgery. Curr Opin Otolaryngol Head Neck Surg. 2021 Oct 1;29(5):385-390.

3. What kind of CI used the patients? Gilbert ML, Deroche MLD, Jiradejvong P, Chan Barrett K, Limb CJ. Cochlear Implant Compression Optimization for Musical Sound Quality in MED-EL Users. Ear Hear. 2022 May/Jun;43(3):862-873

Author Response

Dear reviewer, thanks for the comments. Regarding the use of the Kolmogorov-Smirnov test we had some incertitude. Continuous variables were reported as median (interquartile range, IQR) as appropriate for data with skewed distribution. For what concerns data with symmetrical distribution, simulation studies have demonstrated that the median is equal to the mean of the distribution for large samples [Hozo SP, Djulbegovic B, Hozo I. Estimating the mean and variance from the median, range, and the size of a sample. BMC Med Res Methodol 2005;5:13]. The IQR provides information about the shape of the distribution and allows for identifying symmetrical and skewed distributions. Furthermore, normality tests have several limitations. Even if it is not the case, the pre-test procedure for normality assessment could inflate the type I error rate. Simulation studies recently showed that the pre-test approach for normality can be misleading and suggest applying a non-parametric test for comparison between groups avoiding pre-test procedures [Schucany WR, Tony Ng HK. Preliminary goodness-of-fit tests for normality do not validate the one-sample Student t. Commun Stat-Theory Methods 2006;35:2275–86]. We described the kind of surgical approach applied in the material and method section. Finally, the CI companies were equally distributed between the four majors available.

Round 2

Reviewer 1 Report

A so called "clinical sensor" is different from a physical sensor for clinical applications. So far as I understand the is focussed on physical sensors. The manuscript is interesting, but should be submitted in a rehablitative journal